# Machine Learning-Based Relay Selection for Secure Transmission in Multi-Hop DF Relay Networks

**Tien-Tung Nguyen** [1,2] , **Jong-Ho Lee** [3] , **Minh-Tuan Nguyen** [1] and **Yong-Hwa Kim** [1,*]

1   Department of Electronic Engineering, Myongji University, Yongin 17058, Korea
2   Telecommunication Division, Industrial University of Ho Chi Minh City, Ho Chi Minh City 700000, Vietnam
3   School of Electronic Engineering, Soongsil University, Seoul 06978, Korea
*   Correspondence: yongkim@mju.ac.kr; Tel.: +82-31-330-6370

**Abstract:** A relay selection method is proposed for physical-layer security in multi-hop decode-and-forward (DF) relaying systems. In the proposed method, cooperative relays are selected to maximize the achievable secrecy rates under DF-relaying constraints by the classification method. Artificial neural networks (ANNs), which are used for machine learning, are applied to classify the set of cooperative relays based on the channel state information of all nodes. Simulation results show that the proposed method can achieve near-optimal performance for an exhaustive search method for all combinations of relay selection, while computation time are reduced significantly. Furthermore, the proposed method outperforms the best relay selection method, in which the best relay in terms of secrecy performance is selected among active ones.

**Keywords:** machine learning; physical layer security; multi-classification; relaying network; ANN

## 1. Introduction

Security for wireless communication networks has become a crucial issue because of the broadcast nature of wireless channels. Unauthorized nodes can easily overhear the confidential information of authorized nodes. Secure techniques utilize secret keys that deploy to the upper layers of wireless networks but require complex algorithms [1,2]. By exploiting the physical characteristics of wireless channels, physical-layer security has been regarded as a promising technique to enhance secure communications [3].

For two-hop systems, cooperative networks assisted by relay nodes have been widely investigated to improve the secrecy performance of the systems [4–9]. For the physical-layer security of cooperative relays, a node selection method has been proposed for amplify-and-forward and decode-and-forward (DF) relaying in a two-hop system [6,7]. The optimal relay selection has been investigated for cooperative wireless networks with multiple relays [8]. Maximum ratio combining (MRC), distributed selection combining, and distributed switch-and-stay combining schemes have been evaluated for opportunistic relay selection systems [9].

For multi-hop DF networks, the performance of multi-hop cooperative relay network has been analyzed using path selection and DF protocol at every hop [10] and a decentralized scheme has been proposed conducting the relay selection at each hop independently [11]. In addition, several approaches for selecting cluster-heads using interest of interaction among Internet of Thing (IoT) devices, physical proximity, channel quality and energy availability have been proposed in order to improve the performance of multi-hop systems [12,13]. The security problem for multi-hop wireless networks has been considered [14,15]. The geometric programming method was used to solve the transmit power allocation problem where full-duplex relays are deployed for multi-hop relaying systems [14]. A relay selection method was proposed to obtain the highest secrecy rate of the system

for the scenarios of one-node relay and multiple relays at each hop [15]. However, the relay selection method presented previously [15] using exhaustive search requires high computational complexity when the number of relays increases.

In recent years, machine learning technologies have been applied to various fields such as image processing [16], energy management [17], security [18], and economics [19]. Machine learning has received considerable research interest in wireless networks, such as resource management for long-term evolution [20], predicting the best modulation order and coding rate for multiple-input–multiple-output (MIMO) orthogonal frequency-division multiplexing systems [21], channel estimation [22,23], antenna selection in wireless networks [24], and power allocation [25,26]. For physical-layer security, two machine learning methods, support vector machine (SVM) and naïve-Bayes, have been investigated for MIMO multiantenna-eavesdropper wiretap channels by transmit antenna selection [27].

In this paper, a relay selection problem was considered to maximize the achievable secrecy rate in a cooperative DF multi-hop network with the presence of an eavesdropper. Here, an artificial neural network (ANN) was used to determine the activation of cooperative relays. The proposed ANN model is trained using the training dataset, where the channel state information (CSI) of all nodes is the input, and the corresponding index for the activation of cooperative relays is the output. The effects of the different number of relays, as well as the positions of eavesdroppers, on the secrecy performance of the considered system were investigated. Simulation results show that the secrecy rate performance achieved by the proposed scheme is almost the same as that achieved by an exhaustive search for all combinations of relay selection. Furthermore, the secrecy performance of the proposed method is better than those of selecting the best relay. By using an offline-trained model, almost all the burden of the algorithm complexity is performed during the training stage. Hence, the complexity only depends on the classifying stage. The rest of this work is organized as follows. In Section 2, the system model is introduced, and the relay selection problem is formulated. In Section 3, details on steps of training data generation are provided, and an ANN model is obtained from the training data. The performance of proposed ANN is evaluated, and the results of different transmission schemes are compared in term of the secrecy rate in Section 4. Finally, the conclusion is presented in Section 5.

Notations: Vectors are noted by boldface small letters, and boldface capital letters are defined as matrices; $\mathbf{E}\{.\}$ is denoted as the expectation operator. $\mathbb{R}^{Lx1}$ represents the vector space of all $Lx1$ real matrices.

## 2. System Model

A wireless relaying network is considered that consists of one source node $S$, one destination node $D$, DF trusted relay nodes $\{R_n | 1 \leq n \leq N_r\}$, and an eavesdropper node, $E$, as shown in Figure 1. All nodes are assumed to be equipped with a single antenna, and operate in half-duplex mode, and there exists a direct link from $S$ to $D$. Next, two transmission schemes, namely cooperative transmission (CT) and two-hop transmission, are expressed.

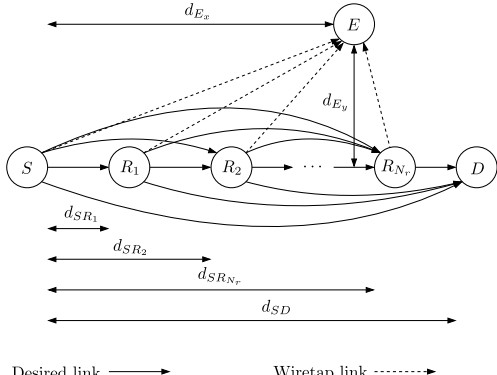

**Figure 1.** System model.

### 2.1. Cooperative Transmission Scheme

In the design, multi-hop relaying transmission is employed. Hence, information transmission between $S$ and $D$ through $N_r$ relays is done during $(N_r + 1)$ time slots. In the first time slot, $S$ uses transmit power $P_0$ to transmit information signal $s$ to all receivers. In this case, it is assumed that $P_0 = P/2$. Therefore, the received signals at $D$, the $n$th relay, and the eavesdropper can be, respectively, given as

$$y_D(0) = \sqrt{P_0}h_{SD}s + z_D(0), \tag{1}$$

$$y_{R_n}(0) = \sqrt{P_0}h_{SR_n}s + z_{R_n}(0), \tag{2}$$

$$y_E(0) = \sqrt{P_0}h_{SE}s + z_E(0), \tag{3}$$

where $h_{SD}$ denotes the complex channel gain of the $S - D$ link, and $h_{SR_n}$ and $h_{SE}$ denote the complex channel gains of] the $S$–$n$th relay link and the $S$–eavesdropper link, respectively. $z_D$, $z_{R_n}$, and $z_E$ are additive white Gaussian noise with variance $\delta^2$ at the receivers.

In the next time slots, $S$ communicates with $D$ via the assistance of the relays that correctly decode signal $s$ (called the active relays). It is assumed that $T \leq N_r$ active relays are selected among the total $N_r$ relays to consecutively transmit information to $D$ during $T$ time slots. Hence, the received signals at $D$, $k$th relay, and eavesdropper $E$ in the $m$th time slot, can be, respectively, shown as

$$y_D(m) = \sqrt{P_m}\left(h_{R_mD}\right)s + z_D(m), \tag{4}$$

$$y_{R_k}(m) = \sqrt{P_m}\left(h_{R_mR_k}\right)s + z_{R_k}(m), \tag{5}$$

$$y_E(m) = \sqrt{P_m}\left(h_{R_mE}\right)s + z_E(m), \tag{6}$$

where $m = 1, 2, \ldots, T; k = m + 1, m + 2, \ldots, T$. $P_m$ is the transmit power of the $m$th active relay, $h_{R_mD}$, $h_{R_mR_k}$ and $h_{R_mE}$ are the complex channel gains of the $m$th relay–$D$ link, the $m$th relay–$k$th relay link, and the $m$th relay–$E$ link, respectively.

It is assumed that the transmit power of each active relay is equal to $P_1/T$, where $P_1$ is the total transmit power of all active relays and it is assumed that $P_1 = P/2$. In addition, all receivers are assumed to use the MRC technique for processing the received signals. Therefore, the rates at $D$, the eavesdropper $E$, and the $m$th active relay with their received signals during $T + 1$ time slots can be computed as

$$\Gamma_D = \frac{1}{(T+1)}\log_2\left(1 + \alpha_{S,D}P_0 + \alpha_{R,D}\left(P_1/T\right)\right), \tag{7}$$

$$\Gamma_E = \frac{1}{(T+1)}\log_2\left(1 + \alpha_{S,E}P_0 + \alpha_{R,E}\left(P_1/T\right)\right), \tag{8}$$

$$\Gamma_{R_m} = \begin{cases} \log_2\left(1 + \alpha_{S,R_m}P_0\right), & m = 1 \\ \frac{1}{m}\log_2\left(1 + \alpha_{S,R_m}P_0 + \alpha_{R_m}\left(\frac{P_1}{T}\right)\right), & m = 2, 3, \ldots, T \end{cases}, \tag{9}$$

respectively, where $\alpha_{S,R_m} = |h_{SR_m}|^2/\delta^2$, $\alpha_{R,D} = \sum_{m=1}^{T}|h_{R_mD}|^2/\delta^2$, $\alpha_{R_m} = \sum_{t=1}^{m-1}|h_{R_t,R_m}|^2/\delta^2$, and $\alpha_{R,E} = \sum_{m=1}^{T}|h_{R_mE}|^2/\delta^2$. Then, the achievable secrecy rate of the considered system can be calculated as

$$\Gamma_{ct} = \max\left\{\Gamma_D - \Gamma_E, 0\right\}, \tag{10}$$

where $\Gamma_D$ and $\Gamma_E$ are given in Equations (7) and (8), respectively.

To guarantee that the above scenario is feasible, it is necessary to check that the active relays can correctly decode the signal from $S$. This is referred to as a "DF relaying constraint", such as $\{\Gamma_{R_m} \geq \Gamma_{th}|m = 1, 2, \ldots, T\}$, where $\Gamma_{th}$ is the rate threshold.

When the DF-relaying constraint is not satisfied, the secrecy rate in Equation (10) cannot be achieved. In the case where $T$ relays among $N_r$ relays are activated, one can consider $\begin{pmatrix} T \\ N_r \end{pmatrix}$ scenarios. Considering that $1 \leq T \leq N_r$, the number of possible relay selection scenarios is $\sum_{T=1}^{N_r} \begin{pmatrix} T \\ N_r \end{pmatrix} = \sum_{T=1}^{N_r} \frac{N_r!}{(N_r-T)!} = (2^{N_r} - 1)$, where $T!$ denotes the factorial of a non-negative integer $T$. For each scenario, the feasibility of this scenario is checked by using the DF-relaying constraint and the secrecy rate is computed as in Equation (10). The scenario providing the highest secrecy rate is the optimal relay (or hop) selection.

Let the highest secrecy rate in this CT scheme be denoted as $\Gamma_{ct}$, given in Equation (10). Furthermore, $T = 0$ indicates that the direct transmission (DT) scheme is used. In this scheme, the rates at $D$, and the eavesdropper $E$, respectively, can be computed as

$$\Gamma_D^{dt} = \log_2\left(1 + \alpha_{S,D} P\right), \tag{11}$$

$$\Gamma_E^{dt} = \log_2\left(1 + \alpha_{S,E} P\right). \tag{12}$$

Then, the achievable secrecy rate of the considered system can be calculated as

$$\Gamma_{dt} = \max\left\{\Gamma_D^{dt} - \Gamma_E^{dt}, 0\right\}, \tag{13}$$

In the no transmission (NT) scheme, the achievable secrecy rate $\Gamma_{nt} = 0$. The goal is to find one case among all possible relay selection scenarios to obtain the highest achievable secrecy rate of the system. Then, the problem of relay selection maximizing the achievable secrecy rate in this multi-hop DF relay network can be formulated as

$$\Gamma_s = \max\left\{\Gamma_{nt}, \Gamma_{dt}, \Gamma_{ct}\right\}, \tag{14}$$

where $\Gamma_{dt}$, and $\Gamma_{ct}$ are given in Equations (10), and (13), respectively.

The solution to the problem can be summarized as the following "theoretical algorithm".

- For the DT scheme, compute the secrecy rate $\Gamma_{dt}$ as in Equation (13).

- For the CT scheme, compute the secrecy rate $\Gamma_{ct}$ as in Equation (10) for all of $\sum_{T=1}^{N_r} \begin{pmatrix} T \\ N_r \end{pmatrix}$ cases where $T$ relays are active.

- Applying Equation (14) for all the secrecy rates, compute the corresponding DT and CT schemes to select the highest secrecy rate.

*2.2. Two-Hop Transmission (Best Relay Selection) Scheme*

In this subsection, the two-hop transmission scheme is considered as a baseline. One of the active relays that satisfies the DF constraint condition is selected to assist the source transmitting signal to the destination. In this case, $T = 1$, information transmission occurs in a two-hop manner. In the first hop, the information is transmitted from $S$ to the selected relay, and, in the second hop, the selected relay decodes the received signals and forwards to $D$.

For this scheme, the rates at $D$, the eavesdropper $E$, and the $m$th active relay with their received signals during two time slots can be computed as

$$\Gamma_{m,D}^{2hop} = \frac{1}{2}\log_2\left(1 + \alpha_{S,D} P_0 + \alpha_{R_m,D} P_1\right), \tag{15}$$

$$\Gamma_{m,E}^{2hop} = \frac{1}{2}\log_2\left(1 + \alpha_{S,E} P_0 + \alpha_{R_m,E} P_1\right), \tag{16}$$

$$\Gamma_{R_m}^{2hop} = \frac{1}{2}\log_2\left(1 + \alpha_{S,R_m}P_0\right), \tag{17}$$

where $\alpha_{R_m,D} = |h_{R_mD}|^2/\delta^2$, $\alpha_{S,R_m} = |h_{SR_m}|^2/\delta^2$, $P_1 = P_0 = P/2$. Then, the secrecy rate of the system for this scheme is obtained as

$$\Gamma_{ct}^{2hop} = \max_{m=1,2,...,N_r}\left\{\Gamma_{m,ct}^{2hop}\right\}, \tag{18}$$

where $\Gamma_{m,ct}^{2hop} = \max\left\{\Gamma_{m,D}^{2hop} - \Gamma_{m,E}^{2hop}, 0\right\}$ is the secrecy rate of each active relay. Then, the problem of relay selection maximizing the achievable secrecy rate in this scheme can be formulated as

$$\Gamma_s = \max\left\{\Gamma_{nt}, \Gamma_{dt}, \Gamma_{ct}^{2hop}\right\}, \tag{19}$$

where $\Gamma_{dt}$, and $\Gamma_{ct}^{2hop}$ are given in Equations (13), and (18), respectively.

The results of the two-hop transmission scheme are only simulated as a benchmark for the proposed relay selection by a machine learning method. In this study, it was assumed that a global CSI is available at $S$. In practice, the end-users (i.e., $E$ or $D$) estimate and feed the absolute values of CSIs from $S$ and all relays back to $S$ [27]. When $N_r$ relays exist, each end-user sends $N_r + 1$ absolute values of CSIs for feedback information.

## 3. Machine Learning for Relay Selection

In this section, a machine learning method is introduced to deal with Equation (14) as a multiclass classification problem. First, features are extracted from the CSIs, and the corresponding class label is obtained for a training dataset. After that a machine learning method, such as the use of ANNs, is trained using the training dataset, where the class label is the corresponding index. In the test dataset, the proposed machine learning method predicts the class label for which the DF relay network can obtain the optimal achievable secrecy rate.

### 3.1. Training Data Design

In this subsection, how to create the training dataset by simulation is described.

#### 3.1.1. Generating Input Data

For the training dataset, $L$ CSIs are generated, and real-valued feature vectors are extracted from these CSIs. Then, the feature vectors are normalized. The feature vector generation is presented as follows:

**Step 1**. Generate the $l$th feature vector $\mathbf{d}^l$ containing the features from CSIs obtained by

$$\mathbf{d}^l = \left[\begin{array}{c} \left|h_{SD}^l\right|, \left|h_{SR_1}^l\right|, \ldots, \left|h_{SR_{N_r}}^l\right|, \left|h_{R_1R_2}^l\right|, \ldots, \left|h_{R_{(N_r-1)}R_{N_r}}^l\right|, \\ \left|h_{SE}^l\right|, \left|h_{R_1D}^l\right|, \ldots, \left|h_{R_{N_r}D}^l\right|, \\ \left|h_{R_1E}^l\right|, \ldots, \left|h_{R_{N_r}E}^l\right| \end{array}\right]^T. \tag{20}$$

**Step 2**. Generate $L$ feature vectors for $L$ CSIs using **Step 1**.

**Step 3**. Normalize feature vector $\mathbf{d}^l$ to obtain the normalized vector $\mathbf{z}^l$. The $n$th feature element of normalized vector $\mathbf{z}^l$ can be calculated as

$$z_n^l = \frac{d_n^l - \mathbf{E}\{\mathbf{d}_n\}}{\max(\mathbf{d}_n) - \min(\mathbf{d}_n)}, \tag{21}$$

where $d_n^l$ is the $n$th element of feature vector $\mathbf{d}^l$, $\mathbf{d}_n \in \mathbf{R}^{L\times 1}$ is the vector containing all $L$ samples for the $n$th feature and $\mathbf{E}\{\mathbf{d}_n\}$ is the expectation of $\mathbf{d}_n$.

### 3.1.2. Labeling

The achievable secrecy rate given in Equation (14) is chosen as the key performance indicator (KPI). From each training data sample and KPI, one can easily determine the class label corresponding to the current sample. There exists one transmission mode to be selected in the considered system during communication between $S$ and $D$. The class labels are indices of cases that contain the important information consisting of the transmission mode and the index of the relay selection combinations. In general, the system has $N_r$ relays, and $\Pi = 2^{N_r} + 1$ class labels are employed, where $2^{N_r}$ class labels are for relay selection combinations and one class label is for NT scheme. An example for labeling is presented in Table 1. It is shown that, when the class label has $t = 0$ with the given the CSI, it means that the system is in an NT scheme. When $t = 1$, the DT scheme is selected. When $t = 2$, the system performs in the CT scheme, in which one relay is active and its index is the first relay, and so on.

**Table 1.** Example of labeling for the system with two relays ($N_r = 2$).

| Transmission Schemes | Labels (t) |
|---|---|
| NT | 0 |
| DT | 1 |
| CT (the first relay is active ) | 2 |
| CT (the second relay is active) | 3 |
| CT (both relays are active) | 4 |

### 3.1.3. Constructing the Training Dataset

After generating the input samples and labeling, the input–output pairs are concatenated to create the full training dataset.

$$D_{train} = \left\{ \left( \mathbf{z}^1, t^1 \right), \left( \mathbf{z}^2, t^2 \right), \ldots, \left( \mathbf{z}^L, t^L \right) \right\}, \tag{22}$$

where $t^i$ is the $i$th class label.

### 3.1.4. Network Structure Design

In this subsection, how an ANN classifier can solve the problem is described. Using the labeled training dataset, a trained ANN model is constructed. The input of the model is absolute values of CSIs and the output is important information such as index of the selected relay set, and one of the transmission schemes. Note that the information transmission of the considered system may occur in one of three transmission schemes, namely, DT, CT, and NT schemes. Here, a brief introduction of a neural network is given. The structure of the neural network contains multiple neural nodes (called units) implemented in each hidden layer. Each layer uses a nonlinear function called an "activation function". The most universal choices for the activation function are the sigmoid function and the rectified linear unit (ReLU) function, which can be, respectively, expressed by

$$f_{\text{sigmoid}}(x) = \frac{1}{1 + e^{-x}}, \tag{23}$$

$$f_{\text{ReLU}}(x) = \max(0, x), \tag{24}$$

where $x$ is the argument of the function. The choice of activation function is a crucial part to ensure good performance of ANNs. The sigmoid activation function is the simplest activation function allowing the neurons learn more complex structures in the data [28]. For a long time, the default activation used on neural network has been the Sigmoid activation function. However, one of the biggest problems during training process with sigmoid activation is vanishing gradient, which may prevent the model from learning effectively as the number of layers get bigger. This is the reason why

ReLU activation function is applied to all hidden layers in our experiment, since it helps the model converge faster without making the gradient saturated as with the sigmoid activation function [29,30].

In a multiclass classification case, an activation function is used at the output layer, which can be formulated as

$$f_{\text{Softmax}}(x_i) = \frac{\exp(x_i)}{\sum_{j=1}^{C} \exp(x_j)}, \tag{25}$$

where $C$ is the number of classes, $i, j \in \{1, 2, \ldots, C\}$, and $x_i, x_j$ are scores of the $i$th class and $j$th class, respectively.

In general, these layers are arranged in a chain structure in which each layer is an activation function of the previous layer, to form

$$\mathbf{o} = f(\mathbf{z}, \mathbf{W}) = f^{(k-1)}\left(f^{(k-2)}\left(f^{(k-3)}\left(\ldots f^{(1)}(\mathbf{z})\right)\right)\right), \tag{26}$$

where $\mathbf{o}$, $\mathbf{z}$, and $\mathbf{W}$ denote the output, the input and the weights of the neural model, respectively, and $k$ is the number of layers of the neural model.

As illustrated in Figure 2, a network is designed containing five layers, which takes the absolute values of CSIs; the first hidden layer, the second hidden layer and the third hidden layer consist of $16 * N_r$, $32 * N_r$, and $64 * N_r$ units, respectively. There are $2^{N_r} + 1$ units at the output layer corresponding to $2^{N_r}$ classes containing secrecy rate values of all combinatorial relay selection cases and one class presenting secrecy rate value of NT scheme. A Softmax function is applied to this layer to represent the probability distribution over all classes, and then obtain the best one with the maximum probability value. This class provides the best combinatorial relay selection or NT scheme corresponding to a given CSI.

For any scale network, an ANN model consists of one input layer, $k \geq 1$ hidden layers and one output layer. The number of elements at the input layer is equal to the total CSIs of all nodes, $N_r * (N_r + 5)/2 + 2$, while the number of neurons of the $k$th hidden layer and the output layer are $2^{(k+3)} * N_r$ and $2^{N_r} + 1$, respectively.

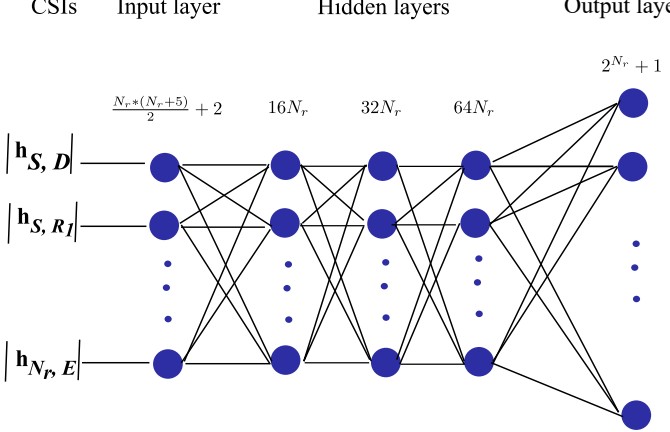

**Figure 2.** ANN model.

### 3.1.5. ANN Training

In this subsection, the selection of appropriate parameters to train the ANN is described. In total, 650,000 samples were generated for training data (i.e., $L = 650{,}000$). The training dataset was split into a training set and validation set. The training set was used to train the network parameters and the validation set was used to evaluate the trained model. As designed in the ANN structure above, cross entropy can be utilized as the loss function for the ANN model. Hence, the loss function for each $i$th sample input $\mathbf{z}^{(i)}$ is formulated as

$$Loss\left(t^{(i)}, o\left(\mathbf{z}^{(i)}, \mathbf{W}, \mathbf{b}\right)\right) = -\log\left(o\left(\mathbf{z}^{(i)}, \mathbf{W}, \mathbf{b}\right)\right) \tag{27}$$

where $t^{(i)}$ is the label representing the best transmission case that provides the highest secrecy rate among $2^{N_r} + 1$ possible cases of the system, and $o^{(i)}\left(\mathbf{z}^{(i)}, \mathbf{W}, \mathbf{b}\right)$ is the output that is predicted by the ANN for $t^{(i)}$ with weight values $\mathbf{W}$ and bias values $\mathbf{b}$. The target of the training process is to find the suitable parameters $\mathbf{W}$ and $\mathbf{b}$ that minimize the average loss (called the "cost function") of entry training dataset. The cost function is defined as

$$L\left(\Theta\right) = \frac{1}{M}\sum_{i=1}^{M} Loss\left(t^{(i)}, o\left(\mathbf{z}^{(i)}, \mathbf{W}, \mathbf{b}\right)\right), \tag{28}$$

where the set $\Theta = \{\mathbf{W}, \mathbf{b}\}$ contains every training parameter of the ANN model. Every parameter is generally updated iteratively using the gradient descent methods. At each iteration, every parameter is updated simultaneously as

$$\Theta^{m+1} = \Theta^m - \eta \nabla_\Theta L\left(\Theta\right), \tag{29}$$

where $\nabla_\Theta$ represents as the gradient operator with respect to $\Theta$, $\eta$ is the learning rate, and $m$ is the iteration number. To optimize the cost function, many gradient descent methods such as Adam, AdaGrad, and AdaDelta optimizers [31–33] are used for updating the network parameters. Based on adaptively changing the learning rate, these optimizers minimize the cost function in a precise manner. In this study, Adam optimization algorithm was applied to the proposed ANN model, because it requires only the first-order gradients to be calculated, thus reducing computational complexity [31]. In addition, to reduce overfit in training, the dropout technique in [34] is applied to ANN model; values of the dropout can be selected in the range 10–90%. However, too large value may result in a slow training and underfitting issue, while too small value may not produce enough dropout to prevent overfitting. Thus, after carefully checking each value of dropout to performance of ANN model, we chose 10% as dropout value, for which the proposed ANN model performs well.

Once the parameters $\mathbf{W}$ and $\mathbf{b}$ are obtained after the training process, the ANN is configured and can calculate the highest secrecy rate of the considered system corresponding to new input vectors $\mathbf{z}$. This means that, any time the channel realizations change, the optimal secrecy rate is updated by feeding the new $\mathbf{z}$ to the trained ANN, without any need to solve the problem defined in Equation (14).

**Remark 1.** *Once the ANN model is trained, the parameters of the trained model can be used at least until the statistical characteristics (such as the probability distribution of complex gain of each channel) of channels change [35,36]. In that case, it is necessary to collect new CSIs from the channels to create the classifying model for the new channel conditions.*

## 4. Numerical Experiments

The effectiveness of our proposed method was evaluated on the optimal achievable secrecy rate. To benchmark, the results of the DT and the two-hop transmission schemes were compared with those of the proposed relay selection method (CT scheme). In addition, for the machine learning method, the performance of ANN model was compared with that of the SVM model.

In the following, distances are denoted between $S - D$, $S - R_n$, $S - E$, $R_n - D$, and $R_n - E$ as $d_{SD}, d_{SR_n}, d_{SE}, d_{R_nD}$, and $d_{R_nE}$, respectively. It was assumed that positions of $S$, $R_n$, and $D$ are in a line as in a previously study [15]. All channels were assumed to experience identical and independent distributed (i.i.d) Rayleigh fadings.

The overall transmit power, $P = 30$ dBm, the noise power $\delta^2 = -30$ dBm, the pathloss exponent $c = 3.5$, and the threshold rate $\Gamma_{th} = 0.1$ (bits/Hz/s) were set. It was assumed that $S$ and $D$ are placed at (0 m, 0 m) and (0 m, 120 m), respectively, or $d_{SD} = 120$ m. All relay nodes were located between $S$ and $D$, and the distance of relay nodes was $\frac{d_{SD}}{N_r+1}$. It was assumed that the positions of the

eavesdropper are changed randomly when $d_{E_x}$ is moved along the parallel line to the line between $S$ and $D$ from 50 m to 180 m and $d_{E_y}$ is located from 5 m to 10 m.

In total, 650,000 samples were generated for training data (i.e., $L = 650,000$). To select the hyper-parameters of the ANN model while avoiding the overfitting problem, 10% of these training data were used randomly for the validation phase. In the ANN model, the batch size was 1024. To ensure the best performance of the model, we trained the model with three different learning rates, 0.01, 0.001, and 0.0001. Based on the results shown in Figure 3, we can see that performance of ANN model with learning rate 0.001 outperforms that of ANN model with learning rates 0.01 and 0.0001 for a six-relay-based system. Hence, we selected the initial learning rate to be 0.001. After obtaining trained ANN model, 10,000 new samples were generated for test data to evaluate the performance of such a model.

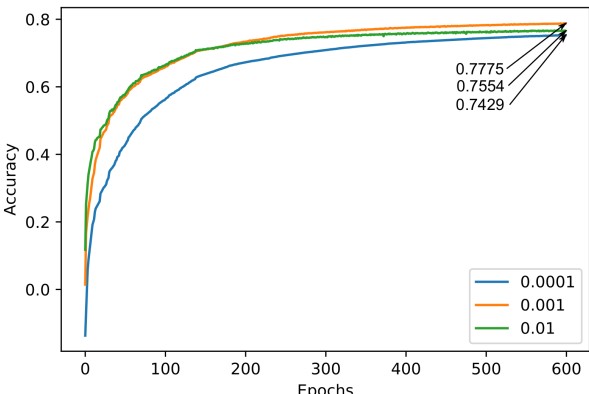

**Figure 3.** Learning rate selection.

Figure 4 depicts the convergence over each training epoch on the training and validation set for three different models corresponding to the number of relays. It can be seen that all lines of accuracies of both the training dataset and validation dataset increase steadily after each epoch. Moreover, the gap between the line of training and the line of validation for each model is minimal, meaning that there is no overfitting problem. In addition, the performance of the ANN method is inversely proportional to the number of relays or input size. Clearly, the accuracy of the ANN model is 95.51% and only 77.75% with two-relay-based system and six-relay-based system, respectively.

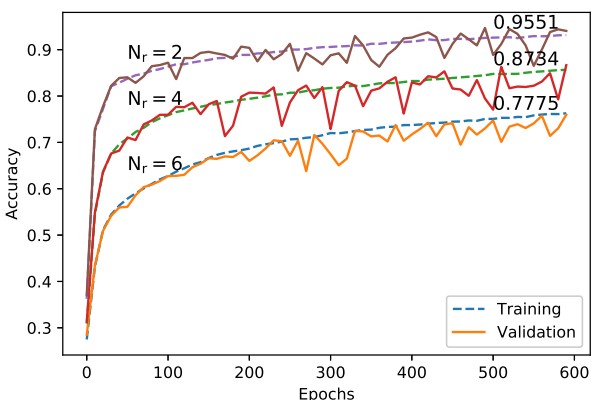

**Figure 4.** Performance converenge versus epochs.

Figure 5 illustrates the achievable secrecy rate changing as a function of number of relays for positions of the eavesdropper corresponding to the horizontal axis $d_{E_x} = 120$ m and $d_{E_x} = 160$ m. In addition, this figure provides a comparison of the exhaustive search method and ANN machine

learning method corresponding to different values of $N_r$. The achievable secrecy rates of both the ANN-based system and the exhaustive search-based system increase when the number of relays $N_r$ increases. In addition, the secrecy performance of the ANN-based system is the same as that of the exhaustive search-based system with $N_r < 4$; however, the secrecy performance of the ANN-based system decreases when the number of relays increases ($N_r \geq 4$). The results also show that, with a given $S - D$ distance, the secrecy performance of the system can be improved by increasing the number of hops.

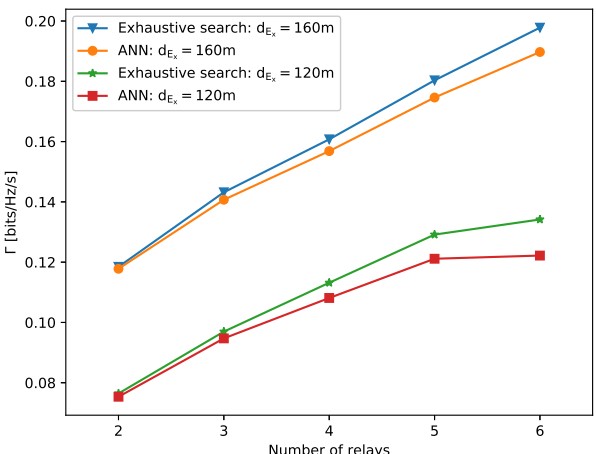

**Figure 5.** The achievable secrecy rate for different number of relays.

Figure 6 presents a comparison of the performances between the ANN method and SVM method and exhaustive search for change of eavesdropper position with different values of $N_r$. Clearly, the position of the eavesdropper has a direct effect on the performance of the machine learning methods. When the eavesdropper is located near the source ($d_{Ex} < 100$ m), the effectiveness of machine learning approach is significant, and the performance of SVM methods are reduced when the position of the eavesdropper is far from the source ($d_{Ex} > 100$ m). In addition, the effectiveness of the ANN method is greater than that of the SVM method for all cases. Moreover, when the network becomes more complex with a greater number of relays, the performance of SVM method drops significantly while the results of ANN method is close to that of the exhaustive search method.

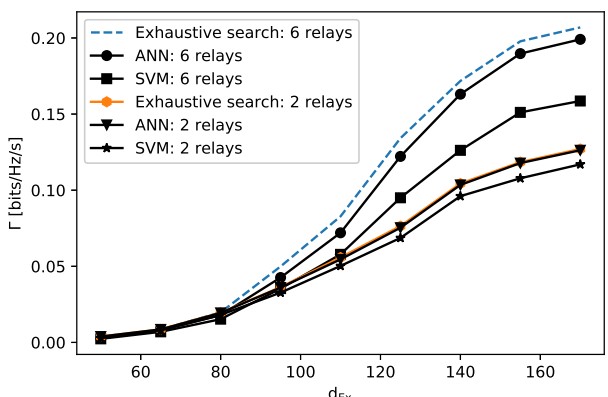

**Figure 6.** Comparison of performance of ANN method with SVM method.

A comparison of three transmission schemes, i.e, the CT scheme, best relay selection (two-transmission) scheme, and DT scheme, is plotted in Figure 7. The secrecy performance of the CT scheme is always much better than that of the best relay selection (two-hop transmission)

scheme and DT scheme. The performance of ANN method decreases when the number of relays of the network becomes larger, but it is always near the optimal values compared with the performance of the best relay selection scheme. This demonstrates the potential of the proposed scheme to be implemented in multi-hop networks. Moreover, the results presented in Figure 7 confirm again the effectiveness of our proposed machine learning method.

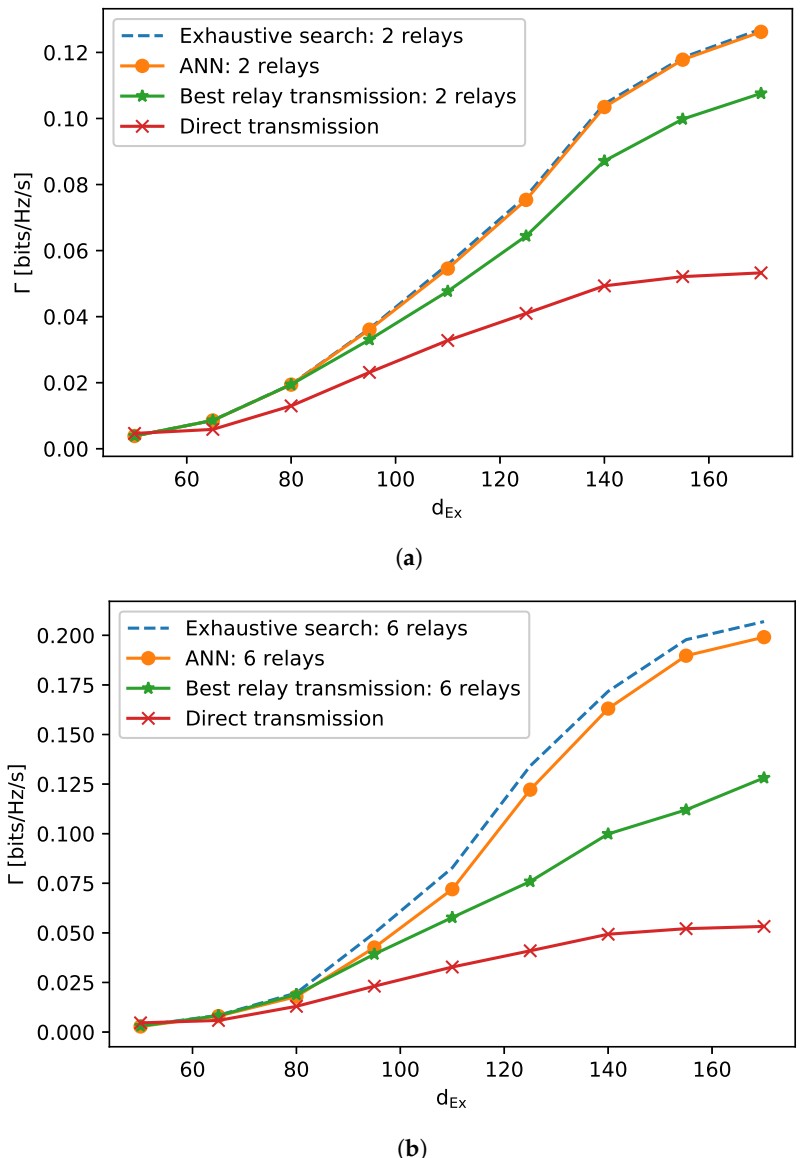

(a)

(b)

**Figure 7.** Comparison of different transmission schemes.

To provide a fair comparison, the exhaustive search algorithm and the proposed machine learning algorithm were implemented in the same platform in Python. Table 2 displays the averaged computational time per sample for both algorithms. The running time of the proposed algorithm outperforms that of exhaustive search algorithm for all values of the number of relays. Furthermore, the computational efficiency of the ANN algorithm is insensitive, whereas that of the exhaustive search algorithm is significantly changed with the increase of relay numbers.

**Table 2.** Comparison of computation time (s).

| Number of Relays | Exhaustive Search | ANN (Test Phase) |
| --- | --- | --- |
| $N_r = 2$ | 0.00286 | 0.00009 |
| $N_r = 4$ | 0.00648 | 0.00018 |
| $N_r = 6$ | 0.03498 | 0.00030 |

## 5. Conclusions

In this work, it was demonstrated that the problem of determining the optimal achievable secrecy rate by selecting active relays for a multi-hop network with DF-relaying constraints can be overcome by using a machine-learning-based method (i.e., using an ANN). First, the simulation results indicate that the proposed method can achieve the secrecy performance of the considered system at optimal values as in the exhaustive search method while the computation time is significantly reduced. Second, increasing the number of hops can enhance the security of the system. Finally, the secrecy performance of the proposed relay selection scheme outperforms that of the two-hop transmission scheme and DT scheme. Moreover, it is hoped that, when applying realistic wireless channels to the simulated data-based model, it will only be necessary to retrain or make minor adjustments without building a new model from scratch. In addition, this study provides insights into the research of new machine learning methods for physical-layer security in wireless cooperative networks. In the future, the proposed model will be applied to a large scale network with multiple eavesdroppers. Moreover, CSIs of eavesdropper not known at the source will be considered.

**Author Contributions:** Y.-H.K. and J.-H.L. conceived of the presented idea. T.-T.N. and N.M.-T. developed the model and performed the computation. All authors discussed the results and contributed to the final manuscript.

**Funding:** This research was supported in part by the Basic Science Research Program through the National Research Foundation of Korea (NRF) funded by the Ministry of Science and ICT (NRF-2017R1C1B1012259), and in part by the Korea Institute of Energy Technology Evaluation and Planning (KETEP) and the Ministry of Trade, Industry & Energy (MOTIE) of the Republic of Korea (No. 17-02-N0202-04).

**Conflicts of Interest:** The authors declare no conflict of interest.

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
