# Peer review of "Machine Learning-Based Relay Selection for Secure Transmission in Multi-Hop DF Relay Networks"

_electronics, doi:10.3390/electronics8090949_

Round 1

Reviewer 1 Report

This paper presents ‘Machine learning-based relay selection for secure transmission in multi-hop DF relay networks’. The idea is interesting.  However, I have some questions and suggestions:

1.      In the abstract, the word ‘best’ is mentioned. It would be good if you could quantify it.

2.      Figure 1 is not very clear. I would suggest redrawing it, please.

3.      Why sigmoid and ReLu were used. It is not enough to mention ‘The most universal choices for the activation function are …’ Please provide a concrete reason.  Additionally why Softmax for the output?

4.      For the sentence ‘machine learning technologies have been applied to various fields from image processing to economics’ only one reference i.e., [13] is not enough. Please provide some more references such as 10.1109/SYSCON.2017.7934803; https://doi.org/10.1007/978-3-030-01057-7_71;

5.      Please double-check the Figure 2. You do not need to mention output Layer and Softmax Layer separately. Please just write Output layer.

6.      How the dropout is calculated? Please explain.

7.      The learning rate 0.001 which is very low. It would be interesting if the author check for other learning rates.

8.      In the ANN training section, could please mention the number of data samples as inputs (training set) to the proposed model.  

This work is interesting but needs to address the above comments.

Author Response

We thank you for your constructive comments and suggestions. We have revised our paper taking into account all of your and other reviewers’ comments/suggestions. We hope you will be satisfied with our revision.

Reviewer 2 Report

The authors focus their study in multi-hop decode and forward relay networks, and they use machine learning techniques to classify the set of cooperative relays based on the channel state information of all nodes. The paper is overall well-written and structured, as well as easy to follow the theoretical provided analysis. The system model formulation and the theoretical analysis are concrete and correct. The authors should address the following comments to strengthen the quality of presentation of their manuscript. 1) The literature review needs to be updated to show the latest advances in the study of the multi-hop decode and forward relay networks, e.g., Senanayake, R., Atapattu, S., Evans, J. S., & Smith, P. J. (2018). Decentralized relay selection in multi-user multihop decode-and-forward relay networks. IEEE Transactions on Wireless Communications, 17(5), 3313-3326, Bhatnagar, Manav R. "Performance analysis of a path selection scheme in multi-hop decode-and-forward protocol." IEEE Communications Letters16, no. 12 (2012): 1980-1983. 2) Except for classifying the set of cooperative relays based on the channel state information of all nodes, other methods have been used in the literature, i.e., nodes’ physical conditions, e.g., Tsiropoulou, Eirini Eleni, Giorgos Mitsis, and Symeon Papavassiliou. "Interest-aware energy collection & resource management in machine to machine communications." Ad Hoc Networks 68 (2018): 48-57, or nodes’ communication interest and context-aware-based communication, e.g., Tsiropoulou, Eirini Eleni, Surya Teja Paruchuri, and John S. Baras. "Interest, energy and physical-aware coalition formation and resource allocation in smart IoT applications." In 2017 51st Annual Conference on Information Sciences and Systems (CISS), pp. 1-6. IEEE, 2017. The authors should update the provided related work to show the holistic approach of the examined problem. 3) In the presented framework, it is not clear what is the signaling overhead imposed to the end-users. 4) The authors should provide a sketch for a large scale implementation scenario. 5) Based on the later comment, the authors should (at least) discuss, if not prove, what is the implementation cost and the computation complexity of this approach in a large scale network. 6) Minor: the usage of the English language needs to be improved; especially check grammar and syntax errors.

Author Response

(The authors gave the same response as above.)

Round 2

Reviewer 1 Report

Paper is revised carefully. I would recommend this paper for publication in its present form. 

Author Response

We thank you for your constructive comments.

Reviewer 2 Report

The authors have addressed all the reviewer's comments.

Author Response

(The authors gave the same response as above.)
